# The ClassyFarm System in Tuscan Beef Cattle Farms and the Association between Animal Welfare Level and Productive Performance

**DOI:** 10.3390/ani12151924

**Published:** 2022-07-28

**Authors:** Francesco Mariottini, Lorella Giuliotti, Marta Gracci, Maria Novella Benvenuti, Federica Salari, Luca Arzilli, Mina Martini, Cristina Roncoroni, Giovanni Brajon

**Affiliations:** 1Istituto Zooprofilattico Sperimentale del Lazio e Della Toscana “M. Aleandri”, Via Castelpulci 43, 50018 Florence, Italy; francescomariottini93@gmail.com (F.M.); giovanni.brajon@izslt.it (G.B.); 2Department of Veterinary Science, Università di Pisa, Viale delle Piagge 2, 56124 Pisa, Italy; martagracci@hotmail.it (M.G.); novella.benvenuti@unipi.it (M.N.B.); federica.salari@unipi.it (F.S.); mina.martini@unipi.it (M.M.); 3Associazione Regionale Allevatori della Toscana, Piazza Eugenio Artom 12, 50127 Florence, Italy; larzilli56@gmail.com; 4Istituto Zooprofilattico Sperimentale del Lazio e Della Toscana “M. Aleandri”, Via Appia Nuova 1411, 00178 Rome, Italy; cristina.roncoroni@izslt.it

**Keywords:** beef cattle, farm animal welfare, ClassyFarm, productive parameters

## Abstract

**Simple Summary:**

Interest in animal welfare has increased due to the growing ethical sensitivity of consumers and the awareness of its impact on food security and safety. In addition, respecting a high standard of animal welfare helps in the prevention of transmissible infectious diseases and the control of antimicrobial resistance. As a response to these needs, in 2018, the Italian Ministry of Health introduced the ClassyFarm system, which categorizes the level of risk arising from farm animal welfare and provides a specific checklist. This paper investigates the relationship between animal welfare and productive parameters on beef cattle farms. Animal welfare was assessed using the ClassyFarm system checklist for beef cattle. Our results demonstrated the influence of animal welfare on productive performance, suggesting that respecting high animal welfare levels helps to reach the full growth potential of beef cattle.

**Abstract:**

In 2018, the Italian Ministry of Health introduced the ClassyFarm system in order to categorize the level of risk related to animal welfare. The ClassyFarm checklist for beef cattle is divided into four areas: Areas A “Farm management and personnel”; B “Structures and equipment”; C “Animal-based measures”; and “Emergency plan and alert system”. Answers contribute to the final Animal Welfare Score (AWS) and to the score of each area. The aim of this work was to assess the animal welfare level on 10 Tuscan beef cattle farms through the ClassyFarm checklist and to examine the relationship between the level of animal welfare on final weight (FW), carcass weight (CW), weight gain (WG), and average daily gain (ADG). The AWS was divided into four classes, and the scores for each area were divided into three classes. The analysis of variance was applied, and AWS class, sex, and breeding techniques (open and closed cycle) were included in the model. The AWS class and sex had a highly significant influence on all parameters, while the breeding technique did not significantly influence any parameter. Farms classified as excellent presented a higher FW (677.9 kg) than those classified as good and insufficient, and the same trend was found for the ADG. The classes obtained in Areas A and C had a highly significant influence on all the parameters investigated. The classes obtained in Area B significantly influenced FW and WG. In conclusion, the productive response of the animals seemed to benefit from the welfare conditions.

## 1. Introduction

Animal welfare is a priority for European consumers. The European Commission is working on legislation to ensure a high level of animal welfare and on the possibility of food labeling to transmit value throughout the food chain. 

Defining animal welfare is complex because it deals with a great number of aspects that need to be taken into account. For example, Hughes (1976) defined animal welfare as “a state of complete mental and physical health, where the animal is in harmony with its environment” [1]. Fraser (2003) recognized three conceptual contexts for assessing animal welfare: biological functioning, affective state, and natural living [2]. For this reason, only considering productive parameters does not guarantee good animal welfare levels but, at the same time, an insufficient productive performance highlights problems that are likely connected with welfare [3]. 

Over the last twenty years, several methods have been developed to assess animal welfare at the farm level, and the ANI 35 L (Animal needs index) represented the first index method used in organic cattle, pig, and hen farms in Austria [4]. ClassyFarm is the officially recognized Italian system to categorize risks on livestock farms and was developed by the Italian National Centre of Reference for the Animal Welfare (CReNBA), funded by the Ministry of Health (www.classyfarm.it (accessed on 20 July 2020)). ClassyFarm is the first Italian welfare assessment protocol for screening the level of farm welfare. The protocol is aimed at supporting official controls, collecting data, promoting the implementation of welfare level, and providing consumers with information.

The system is based on expert opinions for every species and kind of production. According to EFSA (2012) [5], hazards and welfare promoters are characterized in order to define and prioritize a list of management and housing factors potentially associated with negative or positive welfare outcomes in animals. Additionally, animal-based measures have been considered to assess the level of animal pain and suffering due to the welfare consequences they measure [6].

The ClassyFarm system is essentially based on prevention, as it leads to better collaboration between operators and competent authorities, in line with EU regulations on official controls on animal health, welfare, food safety, and pharmaceuticals [7].

The checklists for animal welfare assessment collect data that can be divided into two groups: those related to resource-based indicators, linked to hazards arising from the management and structures, and those referring to animal-based measures (ABMs). 

Among the first indicators, information on livestock management, facilities, equipment, and microclimatic conditions are fundamental for the assessment of animal welfare. Animal-based measures (ABMs) are directly related to the animal’s experience and their ability to cope with the given environment [5,8,9].

It has become increasingly important to assess ABMs that can be directly measured on the animal (for example cleanliness, lameness, or BCS) or indirectly through on-farm data collection (mortality rate) [10].

Productive parameters are often used as indicators in the evaluation of animal welfare at the farm level [11]; however, the association of environmental characteristics that threaten animal welfare and productivity have been documented, especially in dairy cows [12] and pigs [13,14]. 

The aim of this study was to analyze data on animal welfare on beef cattle farms in Tuscany using the ClassyFarm system and to study the relationship between the animal welfare score and productive performance: final weight (FW), carcass weight (CW), weight gain (WG), and average daily gain (ADG).

## 2. Materials and Methods

### 2.1. Experimental Design

The study was carried out between July 2019 and July 2021 on ten Limousine beef cattle farms, eight of which were located in Mugello, a hilly and mountainous area in the province of Florence, and two in the neighboring province of Arezzo *(*Figure 1). According to the zootechnical register of the National Database of the Italian Ministry of Health, the Limousine breed was the most present in Tuscan farms in the years 2020 and 2021, thanks to its rusticity and its productive performances (https://www.vetinfo.it/j6_statistiche/#/report-pbi/11 (accessed on 8 July 2022)).

The farms included in the study were those that were part of the Tuscan regional project “Bencarni: Animal welfare as a tool for the enhancement of the meat supply chain in Tuscany” (Table 1).

Three farms were open-cycle and purchased calves for fattening, while seven were closed-cycle and beef cows were reared according to the cow–calf line, with animals reared in semi-extensive systems and calves sent for fattening at the age of six months. The fattening phase takes place in housing systems made up of closed or open barns with multiple pens. The flooring mostly consists of concrete floors with bedding. The study involved 919 animals. The animals considered using the checklist were 0–6-month-old calves and fattened over 6 months until slaughter; they were slaughtered when aged between 14 and 20 months. 

#### 2.1.1. Data Collection on Growth Performances

Data regarding slaughtering were provided by the local slaughterhouse and included the following information for each animal: registration number, barn code, date of birth, date of slaughter, FW, CW, and WG, which was calculated by subtracting the birth weight from the live weight of each individual animal [15]. Birth weight was estimated according to Simčič et al. [16]. ADG was calculated by dividing the WG of each animal by the age expressed in days.

#### 2.1.2. Animal Welfare Assessment

Two trained veterinarians performed the visits that were arranged with the farmer. The welfare assessment of the 10 farms included in the study was performed in the first semester of 2021.

Animal welfare was assessed using the checklist of the ClassyFarm system for beef cattle (https://www.salute.gov.it/portale/temi/p2_6.jsp?lingua=italiano&id=5174&area=sanitaAnimale&menu=VAeCF (accessed on 8 July 2022)). The checklist is composed of 71 multiple-choice items, with each one having a different weight according to the level of risk represented for animal welfare and health [6]. In addition, 32 items out of 71 comply with the minimum requirements established by current national and European legislation. 

The observations referring to farming conditions are divided into four areas: A (Farm management and personnel), B (Structures and equipment), C (Animal-based measures), and Emergency plans and alert systems. Table 2 reports the items included in the three Areas.

The assessment of each item has two or three answer options: insufficient and acceptable or insufficient, acceptable, and optimal. The thresholds between the different levels of judgment are identified on the basis of the possibility for animals to meet their biological needs and to enjoy the five freedoms [17] underlying animal welfare (impediment, permitted guarantee, and optimal guarantee).

The set of answers are recorded in an algorithm that returns a final percentage expressing the farm’s AWS [10]. The AWS is calculated on the basis of 50% from Area A and Area B, and 50% from Area C [18]. The AWS and the scores of each area are expressed as percentages. The AWS includes the assessment of the Biosecurity score, while the assessment of the “Emergency plans and alert systems” is not included in the algorithm. The sample size for ABM indicators is reported in Table 3.

At the end of the evaluation, a report is produced, which highlights any critical points. 

#### 2.1.3. Data Analysis

According to the farms’ score distribution obtained, the AWS was divided into four classes: insufficient (score < 60%), sufficient (between 60% and 70%), good (between 71% and 78%), and excellent (>78%). Three farms were “sufficient”, four farms “good”, and three “excellent” (Table 4).

The individual areas (A, B, and C) were also divided into three classes: “insufficient” (score < 60%), “adequate” (between 60% and 75%), and “excellent” (score > 75%). Table 4 shows the welfare class distribution of the farms according to the AWS and individual areas.

Statistical analysis was performed via ANOVA using two different models. In the first, the relationship between FW, CW, WG, ADG, and AWS class was verified. In the second, the relationship between these parameters and the classes resulting in each individual area was tested. In both models, sex and breeding technique (open and closed cycle) were added as variability factors. Data were tested for normality. JMP statistical software was used [19].

## 3. Results

### Animal Welfare Assessment and Growth Performance

Data on the influence of AWS class, sex, and breeding technique (open or closed cycles) on FW, CW, WG, and ADG are shown in Table 5. The statistical analysis showed that AWS class and sex had a highly significant influence on all investigated parameters (*p* < 0.0001), while the breeding technique did not significantly influence any of the parameters.

Table 6 shows the data on the influence of the classes of the specific area involved in the determination of the AWS on the investigated parameters. The welfare classes of Area C “Animal-based measures” had a highly significant influence on all the parameters. The classes obtained in Area B “Structures and equipment” significantly influenced FW and WG.

## 4. Discussion

In terms of the influence of the AWS class on the productive parameters, farms classified as “excellent” presented animals with a significantly higher FW (677.9 kg) than those classified as “good” (622 kg) and “insufficient” (588.1 kg) (Table 5). The same trend was found for the ADG, showing that the high welfare level resulted in full productive potential. In fact, the farms classified as “excellent” showed an ADG of 1.03 kg, while those in the “good” and “sufficient” classes showed 0.96 kg and 0.87 kg, respectively (Table 5). This confirms that a good level of welfare has a positive influence on animal productivity [14]. Productive parameters represent useful indicators of animal welfare in many species, and in bovine, they are well documented in dairy cows [12].

In fact, less stressful situations can thus improve the immune system response and reduce disease susceptibility [20,21], thus improving performance.

The statistical analysis confirmed that sex has a highly significant influence (*p* < 0.0001) on all parameters [22,23].

The breeding technique (closed or open cycle) did not significantly influence any of the parameters investigated, probably because fattening, which is managed in confined similar facilities and according to a standardized feeding scheme [20], hides the effect of the previous breeding phase.

Farm management (Area A) had a highly significant influence on productive performance (Table 6); farms classified as “excellent” in this area presented significantly higher FW and ADG than farms classified as “good”. The lack of significance between the class “excellent” and “insufficient” could be due to the low numbers of the “insufficient” sample (n = 84) compared to the “excellent” one (n = 539).

The report showed that the most frequently observed critical points were: hygiene, cleanliness and management of housing and bedding for cattle aged over six months, and group management. Regarding this latter point, marked inhomogeneity between animals in the same box can cause considerable competition for access to resources [24]. However, potential adverse effects resulting from poor conditions can be contained by the animal’s ability to adapt to the environment. In fact, if the animal responds positively to stress, it can also maintain high production levels [25].

The items related to feeding, which were included in Area A, and which represent the most important factor for achieving an adequate productive performance, were not found to be critical points on any of the farms. However, the checklist does not evaluate the composition of the ration.

Concerning biosecurity, which contributes to the results for Area A, we found a rather low and discontinuous level of biosecurity measures, confirming other studies on dairy cattle [20]. Although farmers are aware of the importance of preventing and controlling the spread of infectious diseases, they lack specific training [18,26,27].

Farms rated as “excellent” in Area B “Structure and equipment” had animals with a higher FW (648.5 kg) and higher WG (608.1 kg) than farms rated as “good” (621.8 kg and 581.3 kg, respectively) (Table 6), thus demonstrating that the characteristics of the facility can have a direct impact on animal health and productivity [28].

The main critical issues that emerged from the report were the type of flooring and the type of housing for animals over six months of age, the size and the correct functioning of the water supply, and the resting area. In beef cattle, competition for space, which leads to a reduction in decubitus time and an increase in energy expenditure, is the cause of a worse production performance in terms of ADG [24].

In addition, the presence of straw compared with concrete slats does improve ADG.

The items that constitute Area C (animal-based measures) did not indicate a specific risk but reflected the health status of the animals. In fact, ABMs describe the level of animal pain and suffering due to the impairment of animal welfare [6]. In the algorithm that determines the final score, the ABM area had a higher impact since the algorithm analyzes the real effects of farming conditions on cattle and therefore contributes to objectively defining an animal’s current welfare condition [10]. We found that farms classified as “excellent” achieved significantly higher FW and ADG (661.3 kg and 0.98 kg) than farms classified as “good” (609.1 kg and 0.91 kg). An important critical factor for beef cattle welfare is the human–animal interaction which scientific evidence links to the positive attitude of the farmer [29,30] and to scrupulous attention to the hygiene of the environment with positive consequences on the disease incidence [30,31]. In Area C, the major critical points concerned the cleanliness of the animals, the presence of skin lesions, and aggressive behavior between animals. The cleanliness of the cattle coats represents an indirect indicator of the management procedures on the farm and of the attention paid by the farmer to the hygienic–sanitary status of facilities and equipment [10]. According to Keane et al. [32], providing additional space leads to improvements in animal cleanliness. It also provides a measure of the comfort of the resting areas.

Another indirect indicator of the adequacy of a facility is the assessment of skin lesions, which highlights the importance of reducing risk factors for cattle safety. In fattening cattle farms, incorrect group management, as emerged from the critical points of Area A, led to increased aggressiveness, which translates into increased stress [24]. In fact, it has been demonstrated that stressful situations affect an animal’s productive performance by negatively acting on the physiological processes of the immune and reproductive systems [33]. Improving animal welfare improves product quality and disease resistance. In fact, a good health status and a low level of stress have a direct impact on animal production [34].

## 5. Conclusions

The farms examined in this study attained scores that guaranteed animal welfare, thus demonstrating the farmers’ sensitivity to this key aspect.

We found that the AWS significantly influenced FW, CW, WG, and ADG. The productive response of the animals appeared to be closely related to the breeding conditions, which in turn is related to a good level of animal welfare. Although the main objective of the ClassyFarm method is not the improvement in production performance, the characterization of farm risk facilitates the identification and correction of critical points, thus enabling animals to express their growth potential and efficiently convert the nutrients supplied. Therefore, as well as satisfying ethical and sustainability requirements, a high animal welfare level also improves farm profitability.

## Figures and Tables

**Figure 1 animals-12-01924-f001:**
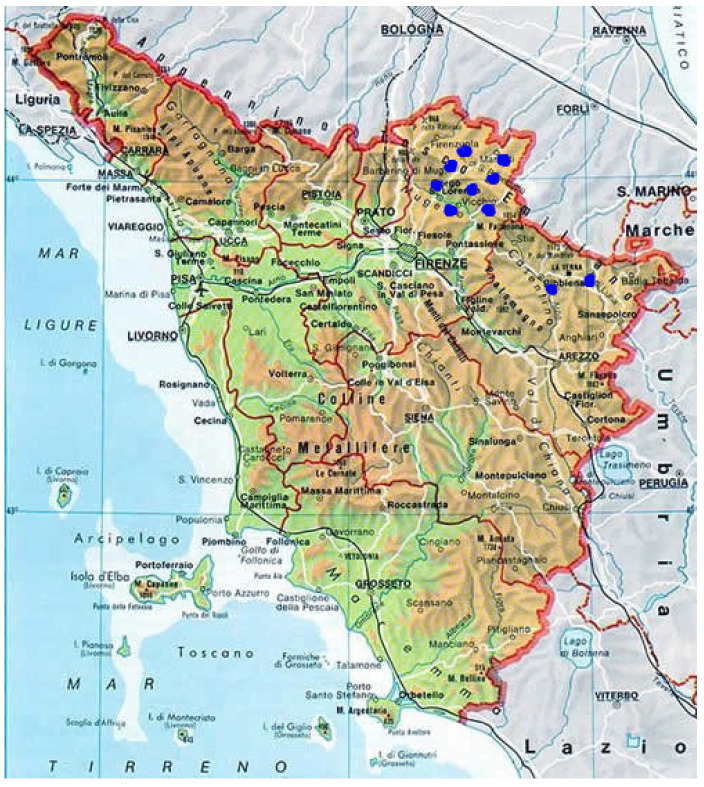
Location of farms.

**Table 1 animals-12-01924-t001:** Characteristics of the farms included in the study.

Farm	Location	Herd Size	Management Practice
1	Arezzo	106	cow–calf line
2	Arezzo	47	cow–calf line
3	Florence	338	fattening
4	Florence	65	fattening
5	Florence	65	cow–calf line
6	Florence	30	fattening
7	Florence	150	cow–calf line
8	Florence	450	cow–calf line
9	Florence	40	cow–calf line
10	Florence	60	cow–calf line

**Table 2 animals-12-01924-t002:** Main items in the three areas.

Area A	Area B	Area C
Number of stockpersons	Management and housing hazards	Agonistic behaviors test
Experience and training of stockpersons	Outdoor shelters	Avoidance distance test
Animal grouping strategy	Housing of animals older than six months	Body condition scoring
Daily inspections of animals	Availability	Animal cleanliness
Treatment of sick or injured animals	Housing system	Skin lesions
Culling	Type of flooring	Lameness
Animal handling	Facilities for sick animals	Respiratory symptoms
Feeding management during the growing and the fattening phase	Temperature, humidity, and ventilation conditions	Mortality rate
Frequency of feed administration	Lighting	Mutilations
Water availability	Air quality and gas concentration	
Number and cleanliness of drinking troughs	Equipment	
Housing and bedding management		
Biosecurity		

**Table 3 animals-12-01924-t003:** Minimum number of animals to observe for ABMs.

Group Dimension	Minimum Number of Animals to Observe for ABMs
<30	All
From 31 to 99	From 30 to 39
From 100 to 199	From 40 to 50
From 200 to 299	From 51 to 55
From 300 to 549	From 55 to 59
From 550 to 1000	From 60 to 63
From 1001 to 3000	From 63 to 65

**Table 4 animals-12-01924-t004:** Distribution of farms (n) in classes according to AWS and the individual area score.

	AWS				Area A			Area B			Area C		
Class	Ins	Suf	Good	Exc	Ins	Adeq	Exc	Ins	Adeq	Exc	Ins	Adeq	Exc
n. farms	0	3	4	3	1	2	7	0	6	4	0	6	4

Ins = insufficient; Suf = sufficient; Good = good; Exc = excellent; Adeq = adequate.

**Table 5 animals-12-01924-t005:** Influence of AWS class, sex, and breeding technique (cycle) on FW, CW, WG, and ADG.

Parameters	AWS				Sex			Cycle			
	Suf N = 301	Good N = 487	Exc N = 131	*p*	F N = 263	M N = 656	*p*	Open N = 509	Close N = 410	*p*	RSME
	Mean (kg)				Mean (kg)			Mean (kg)			
FW	588.1 C	622.0 B	677.8 A	<0.0001	542.1	716.4	<0.0001	634.2	624.4	0.1845	89.493
CW	347.6 C	367.9 B	400.8 A	<0.0001	314.5	429.7	<0.0001	375	369.2	0.1899	53.533
WG	547.5 C	581.5 B	637.3 A	<0.0001	503.1	674.4	<0.0001	593.7	583.9	0.1845	89.493
ADG	0.87 C	0.96 B	1.03 A	<0.0001	0.80	1.11	<0.0001	0.96	0.95	0.6758	0.1503

Suf = sufficient; Good = good; Exc = excellent. Means within the same row with different letters differ significantly (*p* < 0.0001).

**Table 6 animals-12-01924-t006:** Influence of each area on the FW, CW, WG, and ADG.

Parameters	Area A Farm Management and Personnel				Area B Structure and Equipment			Area C Animal-Based Measures			
	Ins	Adeq	Exc	*p*	Adeq	Exc	*p*	Adeq	Exc	*p*	RSME
	n = 84	n = 296	n = 539		n = 807	n = 112		n = 704	n = 215		
	Mean (kg)				Mean (kg)			Mean (kg)			
FW	624.3 AB	628.1 B	653.2 A	0.0071	621.8	648.5	0.0071	609.1	661.3	0.0002	89.239
CW	369.3 AB	371.2 B	386.3 A	0.0072	367.7	383.5	0.0072	360.2	391.1	0.0002	53.384
WG	583.8 AB	587.5 B	612.7 A	0.0071	581.3	608.1	0.0071	568.6	620.8	0.0002	89.239
ADG	0.94 B	0.91 B	0.99 A	<0.0001	0.94	0.96	<0.0001	0.91	0.98	0.0044	0.1501

Ins = insufficient; Adeq = adequate; Exc = excellent. Means within the same row with different letters differ significantly (*p* < 0.001).

## Data Availability

Not applicable.

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
