# Peer review of "The ClassyFarm System in Tuscan Beef Cattle Farms and the Association between Animal Welfare Level and Productive Performance"

_animals, 2022, doi:10.3390/ani12151924_

Round 1
Reviewer 1 Report
General comments:
The manuscript titled “The ClassyFarm System in Tuscan Beef Cattle Farms and the Influence of Animal Welfare on Productive Parameters” investigated the association between animal welfare scores and the productive performance of beef cattle on ten farms in Italy. It seems that the study took too much time to collect data however the manuscript has many flaws that need to be addressed before it is considered for publication. Many important pieces of information are missing such as the list and definition of welfare indicators measured in the study. There is no mention of what age of animals is assessed for welfare. The authors have provided the link for the on-farm welfare audit “ClassyFarm”; however, it is provided in the Italian language, so I was not able to identify the measured indicators. Therefore, the authors should provide an English version of ClassyFarm as supplementary material.
The materials and methods are missing important information that precludes me from assessing the obtained results and discussion. For example:
Authors should provide detailed information on how they conducted the welfare assessment, and what animal age groups were used for evaluation. Were all farms assessed at the same time? How many observers were included in the assessment? Were observers checked for reliability? Please provide that detailed information.
Authors should indicate the sample size for each welfare indicator assessed in the study and how the sample sizes were calculated.
Authors should indicate how many observers conducted the assessment. Did the observers receive training before the start of the study? What was the background of the observers? Please report the interobserver reliability.
What are the criteria used for the selection of study dairies? Were dairy owners informed of welfare assessments before the start of the study? Did the welfare assessment represent the entire age range of all animals on-farm?
The discussion is not well written. The authors should interpret the conclusions in the context of other relevant studies/current evidence.
Specific comments: Please see attached pdf file.
Thanks!

Reviewer 2 Report
The article covers an interesting subject. The assessment of animal welfare on farm and the use of this assessment not only for internal purposes but to inform controls and other institutions is an approach which Italy is establishing and which could be exemplary for other European member states.
The article should inform the reader – who most often will not know ClassyFarm - more thoroughly about the system. This would also allow for a better understanding and the possibility to discuss the results achieved. Information which should be provided are: the full list of indicators, an overview of the aggregation system, a description of the way the audits are performed (trained auditors? Veterinarians or other profession? Announced visits on farm? Are all farms visited from all animal species? Are the visits carried out once a year?).
The authors should provide their definition of animal welfare (AW) and reflect the results accordingly. One of the results is that the animal based measures correlate with the weight gain. As these animal based measures only cover health issues, and the AW dimensions of “natural behaviour” and “emotional state” (if one chooses an AW relating to Frasers 3 Dimensions concept) are not represented. This should be part of the discussion.
Productivity and AW are not necessarily positively linked. This needs to be discussed and the respective literature should be used and cited. The same applies to biosecurity (one example is that all poulty has to be kept indoors due to avian influenza outbreaks), and antibiotic use.
Some additional information on the farms should be provided: what size (cattle numbers) are they? Is this size representative for all Tuscan cattle farms? Is Limousin a common breed in Tuscany? Why were only Limousin cattle farms selected?
The sample is rather small, why could only so few farms be included into the study? Where the others not willing to participate, would the willingness to participate influence the outcome of the analysis?
The indicators chosen to analyse the effects of animal welfare on productivity are closely related to each other, collinearity should therefore be tested for.
The effect of sex on weight gain is known and does not need a re-assessment. Wouldn’t it have been more useful to carry out the whole analysis separately for male and female animals?
Round 2
Reviewer 1 Report
General comments:
This link that the authors provided still does not show the checklist of welfare indicators used for welfare assessment. Please provide detailed information on each indicator assessed in the study.
The table that the authors provided does not provide sufficient information on the measured items, and there is no definition of answers under each item.
The figure does not show the location of each farm on the map. Please provide more information to discuss what is included in the map.
Extensive editing of the English language and style is required for this manuscript.
Again, you studied the association between the AWS and productive performance not the influence. Please check throughout the manuscript and correct it accordingly.
Specific comments: See attached pdf.

Round 3
Reviewer 1 Report
Thanks authors for answering my comments, however, I can't see any edits in the submitted pdf. Please make sure to address my comments in your final version. Congratulations on your paper! All the best.